# A Comparative Study of the Inhibitory Action of Berberine Derivatives on the Recombinant Protein FtsZ of *E. coli*

**DOI:** 10.3390/ijms24065674

**Published:** 2023-03-16

**Authors:** Angela Di Somma, Carolina Canè, Natalie Paola Rotondo, Maria Maddalena Cavalluzzi, Giovanni Lentini, Angela Duilio

**Affiliations:** 1Department of Chemical Sciences, Università Federico II di Napoli, Vicinale Cupa Cintia 26, 20126 Naples, Italy; 2CEINGE Advanced Biotechnologies, Via Gaetano Salvatore 486, 80131 Naples, Italy; 3Department of Pharmacy–Pharmaceutical Sciences, University of Bari Aldo Moro, Via E. Orabona n. 4, 70126 Bari, Italy

**Keywords:** berberine derivatives, antimicrobial agent, FtsZ, competitive inhibition

## Abstract

Medicinal plants belonging to the genus *Berberis* may be considered an interesting source of drugs to counteract the problem of antimicrobial multiresistance. The important properties associated with this genus are mainly due to the presence of berberine, an alkaloid with a benzyltetrahydroisoquinoline structure. Berberine is active against both Gram-negative and Gram-positive bacteria, influencing DNA duplication, RNA transcription, protein synthesis, and the integrity of the cell surface structure. Countless studies have shown the enhancement of these beneficial effects following the synthesis of different berberine analogues. Recently, a possible interaction between berberine derivatives and the FtsZ protein was predicted through molecular docking simulations. FtsZ is a highly conserved protein essential for the first step of cell division in bacteria. The importance of FtsZ for the growth of numerous bacterial species and its high conservation make it a perfect candidate for the development of broad-spectrum inhibitors. In this work, we investigate the inhibition mechanisms of the recombinant FtsZ of *Escherichia coli* by different *N*-arylmethyl benzodioxolethylamines as berberine simplified analogues appropriately designed to evaluate the effect of structural changes on the interaction with the enzyme. All the compounds determine the inhibition of FtsZ GTPase activity by different mechanisms. The tertiary amine **1c** proved to be the best competitive inhibitor, as it causes a remarkable increase in FtsZ *K*_m_ (at 40 μM) and a drastic reduction in its assembly capabilities. Moreover, a fluorescence spectroscopic analysis carried out on **1c** demonstrated its strong interaction with FtsZ (*K*_d_ = 26.6 nM). The in vitro results were in agreement with docking simulation studies.

## 1. Introduction

Nowadays, bacterial resistance to antibiotics represents a major challenge in the fight against infection, as it reduces therapeutic options, leading to higher medical costs, prolonged hospital stays, and increased mortality [1]. The rapid emergence of resistant bacteria then requires the development of new strategies to increase the efficacy of antibiotics. The phenomenon of resistance is strictly related to the mechanism of action of the antimicrobial agents, and the identification of the specific vital microbial functions they are targeting is pivotal when looking for new pharmacological tools to fight resistant bacteria [2,3].

The most widely studied bacterial target for novel drug development is the cell division machinery [4]. In bacteria, this vital function is accomplished by the divisome, a macromolecular complex that is highly dynamic and is characterized by a time-dependent assembly of specific cell division proteins [5]. The process depends on the FtsZ protein (filamenting temperature-sensitive mutant Z), which is found in almost all bacteria and is essential for their viability [6]. This tubulin homolog protein undergoes GTP-dependent polymerization, assembling into a ring at the future site of bacterial cell division and recruiting the other cell division proteins to complete the division process [7].

As FtsZ is conserved in most bacteria, but is absent in eukaryotes, this protein is considered a good target for the development of a new class of antibiotics able to specifically impair bacteria cell division. FtsZ displays two druggable pockets, the nucleotide-binding site, where GTP or GDP binds, and the C- terminal site, where many already developed inhibitors were shown to interact [8].

Some molecules, such as antimicrobial peptides or natural and synthetic small molecules [9,10], have recently been reported as FtsZ inhibitors preventing bacterial cell division. In most cases, the GTP binding pocket or the H7 helix were found to be the binding targets of these compounds. These antimicrobial agents were demonstrated to interfere with the natural dynamics and functions of FtsZ during the cell cycle, bringing about death in a suicidal manner [11]. However, for different reasons, these inhibitors could not show the specific characteristics needed to move forward to clinical trials [12].

Plants, with their antibacterial properties, might represent a solution to the problem of antibiotic resistance, as they produce many natural defence compounds [13]. Among these, important properties associated with the genus Berberis are due to the presence of berberine, a modified benzyltetrahydroisoquinolinic skeleton alkaloid [14,15]. The antimicrobial properties of berberine have already been proven against a multitude of bacterial species [16]. Due to the presence of planar polyoxygenated cycles and a net positive charge, it can exert its antibacterial activity towards both Gram-positive and Gram-negative bacteria with different mechanisms [17].

In our previous study, molecular docking simulations suggested FtsZ as a possible target of berberine derivatives and analogues prepared according to the structural simplification approach [18]. In particular, **1c**- and **1f**-substituted berberine analogues showed greater potency than their parent compounds [19], indicating that appropriate modifications to the chemical structure of berberine might be instrumental to obtaining molecules with higher antimicrobial properties and FtsZ inhibitory capabilities. In this paper, two new structural berberine simplified analogues have been developed based on an attempt to maximize their interaction with FtsZ. We aimed to study in silico and in vitro the binding properties, inhibitory capabilities, and antimicrobial activities of both new and previously reported compounds with different experimental techniques, including fluorescence measurements and enzymatic and antimicrobial assays.

## 2. Results

In a previous paper, we investigated in silico the interaction of berberine and some of its simplified analogues (**1a**–**f**) with the *E. coli* protein FtsZ by molecular docking simulations (Figure 1). Surprisingly, a different behaviour was observed for these compounds. Berberine and compounds **1a** and **1f** were predicted to interact with a binding site near the C terminus of the protein, whilst the analogues **1b**–**e** were hosted in the protein active site within the GTP-binding pocket, thus suggesting a possible competitive inhibition mechanism on the GTPase activity of the protein [18].

Therefore, to verify the docking predictions, appropriate in vitro assays were carried out. Furthermore, two novel compounds (**2g**,**h**) were designed by merging the structural elements identified in the previous work as responsible for the highest antibacterial activity against *E. coli*—namely, the N-methyl substitution (see compound **1c**) and the presence of a N-4-chlorobenzyl- or N-naphthalenmethyl moiety (see compounds **1g** and **1h**, respectively)—assuming a further activity increase.

### 2.1. Chemistry

Compounds **2g**,**h** (Figure 2) were prepared starting from the secondary amines **1g**,**h**, which were alkylated with methyl iodide, as previously reported [20].

### 2.2. Molecular Docking Analysis

The possible interaction between compounds **1g**–**i**, **2g**,**h**, and the FtsZ protein was first investigated by in silico docking simulation experiments using PatchDock Server and FireDock Server. The *E. coli* FtsZ was modelled with the I-TASSER Webserver, whilst the LigParGen Server was used to obtain the structures of the berberine analogues under evaluation. Three analogues localised within the active site pocket of the protein (**1g**,**h** and **2h**), one compound showed interaction with the C-terminal region of FtsZ (**1i**), while the fifth analogue (**2g**) seemed to bind to a completely different protein region (Figure 1).

In particular, the **1g**,**h** and **2h** analogues were located in a hydrophobic pocket found by the helices H1, H2, and H7 and the hydrophilic pocket surrounded by the glycine-rich loops T1, T3, and T4. These compounds established non-covalent interactions, hydrophobic interactions, and hydrogen bonds with key amino acids of the nucleotide-binding site, Gly 21, Gly 107, and Thr 108 (Table 1). However, a number of different contacts with amino acid side chains were predicted for the three analogues. As reported in Figure 2, the **2h** derivative seemed to place the benzodioxole ring in a completely different position in the active site than the other compounds. It showed a completely flipping arrangement. In this position, the **2h** compound does not contact the Gly19, Gly21, and Asn165, but establishes more interactions with the nucleotide-binding site, increasing the H bonds. A higher number of non-covalent bonds and a π-stacking interaction between Phe182 and the naphthalene ring were also predicted.

The **1i** analogue was predicted to interact with the hydrophobic cleft between the H7 and C-terminal β sheet of the FtsZ protein. Finally, compound **2g** does not bind FtsZ in the active site or C-terminal, but in a totally different region, and it established four hydrophobic interactions, one hydrogen bond with Lys 352, and one non-covalent interaction with Asp 370.

The main interactions predicted for the complexes between the FtsZ protein and the berberine analogues under evaluation are summarised in Table 1. The Gibbs free energy analysis was also calculated using the PRODIGY webserver, yielding negative ΔG values for most of the compounds and suggesting the formation of protein–ligand complexes with comparable stability. Among the analogues, compound **2g** showed the highest ΔG value (−7.07 kcal/mol), suggesting the formation of a less stable complex.

### 2.3. Inhibition of FtsZ Enzymatic Activity by Berberine Analogues

The functional effect of the berberine analogues **1c**,**d**,**f**,**i**, and **1**-**2g**,**h** on FtsZ was investigated by enzymatic assays. A recombinant form of FtsZ was produced in *E. coli*, purified according to [21], and used for binding assays. The GTPase activity of the FtsZ protein was assayed in the presence of the berberine-related compounds at 40 μM concentration and was monitored in comparison with the untreated protein at different GTP concentrations. The kinetic parameters were calculated according to the Michaelis–Menten equation using GraphPad Prism 8 for all the berberine-related compounds. The software returned K_M_ (μM) and V_max_ (Pi released/min) kinetic parameters that indicated the mechanism of inhibition exerted by each compound. The results are reported in Table 2. A clear decrease in the enzymatic activity of FtsZ was observed in the presence of all berberine analogues, demonstrating their ability to inhibit the enzyme.

A careful inspection of the kinetic parameters highlighted the occurrence of different inhibition mechanisms exerted by the berberine-related compounds. A classical competitive inhibition mechanism was demonstrated for compounds **1c**,**d**,**g**,**h**, and **2h**, as they caused an increase in the apparent *K*_M_ values, while the Vmax value remained unchanged (Table 2).

Conversely, the **1f** and **1i** compounds showed a non-competitive inhibition mechanism, displaying a decreased value of Vmax with a constant *K*_M_ value. Surprisingly, the **2g** analogue showed a completely different behaviour compared to the other tested analogues, displaying decreased values of both Vmax and *K*_M_, indicative of an acompetitive inhibitory mechanism.

These observations were in perfect agreement with the structural predictions provided by the docking experiments.

The determination of the inhibition constant *K*_i_ for these compounds was performed to further investigate the mode of inhibition and clarified the potency of the competitive inhibitors. The *K*_i_ values were calculated performing kinetic analyses at different concentrations of competitive inhibitors (20 μM, 40 μM, and 60 μM). The results are shown in Table 3.

The lowest value was calculated for the **1c** analogue (25 µM), suggesting it as the most promising inhibitor of FtsZ.

### 2.4. Inhibition of FtsZ Polymerization in the Presence of Berberine Analogues

Inhibition of the GTPase activity of FtsZ might prevent the polymerization of the protein, which is well known to occur in the presence of GTP. We then investigated the effect of berberine analogues on the polymerization of FtsZ. The recombinant protein was incubated with 150 µM GTP in the absence and in the presence of increasing concentrations of berberine-related compounds. The FtsZ filaments were purified by centrifugation, analysed by SDS-PAGE, and the amount of polymerized protein was determined by densitometric analysis of the corresponding Coomassie-stained gel bands. The results demonstrated that all the compounds were able to impair the ability of FtsZ to polymerize into long filaments, despite their different inhibitory mechanisms. Figure 3 clearly shows that the amount of polymerized FtsZ decreased in a dose-dependent manner as the concentration of the berberine analogues increased.

Among the berberine derivatives, the **1c**,**f**,**i** compounds seemed to exhibit the best inhibitory effect on the polymerization of FtsZ, since they cause a reduction in the polymerized material of 35, 30, and 25%, respectively. However, the **1i** compound exerts the highest inhibitory effect, as the amount of polymerized FtsZ was almost negligible when the inhibitor concentration was 60 µM.

The capability of berberine analogue **1i** to inhibit FtsZ polymerization was also assessed by TEM microscopy, by analysing the FtsZ protofilaments of in the presence of 60 µM **1i** in comparison with the untreated sample. Figure 4 shows that FtsZ forms a thick bundle of GTP-induced protofilaments in the absence of the inhibitor. However, when the measurement was performed in the presence of 60 μM **1i**, no protofilaments could be observed, indicating the ability of the **1i** inhibitor to prevent FtsZ polymerization (Figure 4b).

### 2.5. Binding of ***1c*** Berberine Analogue to the FtsZ Recombinant Protein

As the **1c** compound exhibits both the best inhibitory activity on the GTPase FtsZ and a significant suppression of FtsZ assembly, we were stimulated to confirm these interactions by fluorescence binding assays. FtsZ does not contain tryptophan residues in the sequence, and following excitation at 295 nm, the emission maximum at 327 nm was negligible. Moreover, while berberine shows strong absorbance at both wavelengths, compound **1c** did not display any absorbance at 327 nm, as shown in Figure 5.

Therefore, the intrinsic fluorescence of compound **1c** was monitored at increasing concentrations of recombinant FtsZ. The sample was excited at 295 nm, and the fluorescence emission of the compound **1c** was recorded. Figure 6a shows a clear quenching in the intensity of the fluorescence emission without FtsZ and with an increasing concentration of FtsZ at 327 nm. Data from the fluorescence quenching allowed us to calculate the dissociation constant value *K*_d_ = 26.64 ± 0.8 nM, confirming the formation of a stable complex between compound **1c** and the protein.

### 2.6. Antimicrobial Activity of Berberine Analogues

Finally, we evaluated the in vitro antimicrobial activity of the newly synthesized berberine analogues (**2g**,**h**) on *E. coli* strain. The Minimal Inhibitory Concentrations (MICs) of both compounds were determined as the lowest concentration showing no visible growth after 24 h of incubation. The MIC values for **2g** and **2h** are reported in Table 4, together with the most interesting compounds previously reported [18].

Although compounds **1g** and **2h** are expected to act similarly, they showed different MIC values. The same was observed for compounds **1c**, **1i**, and **1f**, whose antibacterial activity did not reflect the inhibitory effect on FtsZ polymerization. Conceivably, these apparent discrepancies could stem from the differences between in silico/in vitro tests involving isolated protein targets and what happens in a living cell. To reach its target inside the E. coli cell, the drug must cross the fairly hydrophobic cell wall of the Gram-negative bacterium; compounds **1f** and **2h** could get trapped due to their high lipophilicity (log P 6.3 and 5.3, respectively). Conversely, the log P values of compounds **1c** and **1g** (3.7 and 4.0, respectively) is perfectly in agreement with the ideal value of 4 for Gram-negative bacterial uptake [22].The peculiar behaviour of the secondary amine **1i** could be explained not only in terms of borderline lipophilicity, its log P being 3.17, but also as a conformational effect. It is reasonable to hypothesize that the presence of the strong electron-withdrawing nitro group could be responsible for an intramolecular π-stacking interaction between the electron-poor benzyl moiety and the electron-rich benzodioxole, thus forcing the molecule into a folded conformation that would confer peculiar partitioning properties, possibly hampering the access into the bacterial cell.

## 3. Discussion

Plants possess a wide array of bioactive compounds that, due to their high safety, tolerance, availability, and low toxicity, might represent an invaluable source of new antibacterial agents with potential therapeutic effects [23]. Many plant-derived compounds are commonly used for the diagnosis and treatment of various diseases [24].

Isoquinoline alkaloids are a large group of plant-based alkaloids possessing an extensive spectrum of biological properties, which include anti-inflammatory, antitumor, antimicrobial, and antioxidant effects [25,26]. Berberine is a natural plant-derived isoquinoline alkaloid widely known for its therapeutic potential that has traditionally been used to treat microbial infections safely at commonly used doses [27]. However, the therapeutic utility of berberine is compromised due to its undesirable pharmacokinetic properties and its poor water solubility. This limitation could, however, be overcome by appropriate structural modifications on its scaffold, leading to the synthesis of various berberine derivatives with improved pharmacological and pharmacokinetic profiles [28].

In vitro studies showed that berberine targets *Escherichia coli* FtsZ, which is a fundamental protein of the cell division machinery. Berberine inhibits the FtsZ GTPase activity, thus impairing the assembly of the Z-ring and blocking cell division [29]. The highly conserved nature of the FtsZ protein among numerous prokaryotic species provides an opportunity for the development of broad-spectrum antibiotics [30]. Recently, molecular docking simulations predicted the binding of berberine analogues, either in the active site pocket, thus impairing the GTPase activity of the protein by a competitive inhibitory mechanism (**1c**,**d**), or near the C-terminal region of the protein, leaving the active site free for GTP binding (**1f**) [18].

We were then stimulated to investigate the possible interaction with FtsZ of three compounds **1g**,**h**,**i** synthesized in the previous work and two novel compounds (**2g**,**h**) appropriately designed. Molecular docking simulations predicted that all compounds would interact with the protein, although showing different behaviours. Compounds **1c**,**d**,**g**,**h** and **2h** were located in the GTP-binding pocket, establishing several interactions with amino acids of the nucleotide-binding site, while the **1a**,**f**,**i** analogues were oriented towards the C-terminal domain of FtsZ, analogously to berberine. Finally, according to docking predictions, compound **2g** was placed at a completely different site of the protein. Interestingly, this region of FtsZ was reported to be crucial for both self-interaction between FtsZ molecules and binding with other proteins (i.e., ZipA protein) [31].

Enzymatic assays carried out with a recombinant version of FtsZ confirmed the occurrence of different inhibition mechanisms, depending on the different interaction sites. As expected, the **1c**,**d**,**g**,**h**, and **2h** analogues gave origin to a competitive inhibitory mechanism, with the **1c** analogue showing the strongest inhibitory effect, as shown by the twofold increase in the *K*_M_ value and the very low *K*_i_ values. Conversely, compounds **1f** and **1i** showed a clear non-competitive inhibitory mechanism, as demonstrated by the decrease of the V_max_ value. Finally, compound **2g** showed a decrease in both the *K*_M_ and V_max_ values, according to an acompetitive mechanism. The binding of this analogue to a specific region of FtsZ possibly resulted in a modification of the enzyme conformation that reduced the affinity of the substrate with the active site and the efficiency of the enzyme, resulting in a decrease of both kinetic parameters. Inhibition of the FtsZ GTPase activity by the berberine analogues also affected the FtsZ polymerization capabilities in a dose-dependent manner, with **1c**,**f**,**i** showing the highest inhibition properties.

Among the tested compounds, the **1c** analogue showed the most promising results in both inhibiting FtsZ GTPase activity and preventing protein polymerization. Consequently, we investigated in more detail the interaction between compound **1c** and the target protein FtsZ by fluorescence binding. Binding of **1c** to FtsZ resulted in the formation of a very stable complex, with a dissociation constant in the low nanomolar range.

Finally, the antimicrobial properties of the berberine analogues in vivo on *E. coli* cells were also investigated, showing different MIC values among the various compounds.

Based on the results reported in this paper, we might draw some considerations about the relationship between the structure of the berberine analogues and their preferential FtsZ binding sites and inhibitory properties. Scaffolds with either a benzoyl moiety decorated with two methoxy groups or a naphtalen ring, carrying small groups on the nitrogen atom (H, Me, or Et), address the berberine analogues towards the active site of the enzyme (**1c**,**d**,**g**,**h** and **2h**), while larger substituents on either the N atom or the benzyl ring promote the binding at the C-terminus (**1f**,**i**).

Among the analogues interacting with the active site, the presence of a small methyl group on the nitrogen atom leads to a remarkable increase in both inhibitory and antibacterial activities. A similar behaviour of inhibitory activities was observed for the analogues preferentially interacting at the C-terminus (**1f**,**i**). A completely different consideration should be drawn for the berberine analogue **2g** having a Cl atom at R^3^ and a Me group on the N atom. This compound binds FtsZ in a totally different region compared to the other analogues, showing good inhibitory properties by an acompetitive inhibitory mechanism, impairing FtsZ polymerization, and showing a low MIC value. These features make analogues **2g** and **1c** promising compounds that deserves further investigation.

The data reported herein suggest that new berberine derivatives target the bacterial protein FtsZ, demonstrating the potential of the berberine scaffold for chemical optimization in potent FtsZ inhibitors with broad-spectrum antibacterial activity and proposing a novel mechanism of action of new berberine analogues in inhibiting FtsZ.

## 4. Materials and Methods

### 4.1. Chemistry

General procedure for the synthesis of compounds **2g**,**h**.

2-(1,3-Benzodioxol-5-yl)-*N*-[(4-chlorobenzyl]-*N*-methylethan-1-amine hydrochloride (**2g**.HCl).

The procedure adopted for the synthesis of 2-(1,3-benzodioxol-5-yl)-*N*-[(4-chlorobenzyl]-*N*-methylethan-1-amine hydrochloride (**2g**.HCl) is described.

A solution of iodomethane (0.1 mL, 1.59 mmol) in 3 mL of absolute EtOH was added dropwise to a magnetically stirred solution of 2-(1,3-benzodioxol-5-yl)-*N*-[(4-chlorobenzyl]ethan-1-amine (**1g**) (0.383 g, 1.33 mmol) and K_2_CO_3_ (0.367 g, 2.66 mmol) in absolute EtOH (12 mL). The reaction mixture was stirred at room temperature for 24 h and then evaporated under vacuum. The residue was taken up with EtOAc and washed twice with brine. After drying (Na_2_SO_4_), the organic phase was evaporated under vacuum to afford 0.350 g (87%) of a yellowish oil, which was purified by column chromatography (EtOAc/hexane 1:1), giving 0.125 g of a yellowish oil: GC-MS (70 eV) *m*/*z* (%): 303 (M^+^, <1), 125 (100). The corresponding hydrochloride (**2g**.HCl) was obtained by dissolving the free base in 1 mL of 2 M HCl and azeotropically removing water (toluene/abs EtOH). The obtained white solid was recrystallized from CHCl_3_/THF/hexane, giving 0.109 g (24%) of white crystals: mp: 171–173 °C; ^1^H NMR (300 MHz, CD_3_OD): *δ* 2.84 (s, 3H, C*H*_3_N), 3.03 (br t, *J* = 8.4 Hz, 2H, C*H*_2_CH_2_N), 3.30 (br t overlapping CD_3_OD, *J* = 8.4 Hz, 2H, NC*H*_2_CH_2_), 4.38 (br s, 2H*,* C*H*_2_N), 5.92 (s, 2H, OC*H*_2_O), 6.74 (dd, *J* = 7.8, 1.5 Hz, 1H, Ar *H*C-6), 6.76 (s, 1H, Ar *H*C-4), 6.76 (apparent bt, 1H, Ar *H*C-7), 7.48–7.58 (m, 4H, Ar *H*); ^13^C NMR (125 MHz, CD_3_OD): *δ* 29.7 (1C), 38.8 (1C), 56.8 (1C), 58.7 (1C), 101.1 (1C), 108.1 (1C), 108.6 (1C), 121.6 (1C), 128.1 (1C), 129.1 (2C), 129.5 (1C), 132.5 (2C), 136.0 (1C), 146.9 (1C), 148.1 (1C); Anal. Calcd for C_17_H_18_ClNO_2_.HCl.0.3H_2_O: C, 58.97; H, 5.73; N, 4.05; Found: C, 58.98; H, 5.60; N, 4.23.

2-(2*H*-1,3-Benzodioxol-5-yl)-*N*-methyl-*N*-[(naphthalen-1-yl)methyl]ethan-1-amine hydrochloride (**2h**.HCl).

Prepared as described above for **2g**.HCl starting from **1**,**h**.HCl in 67% yield. GC-MS (70 eV) *m*/*z* (%): 184 (M^+^–135, 35), 141 (100).

Data for **2h**.HCl (brown crystals, 50%): mp: 188–190 °C; ^1^H NMR (300 MHz, CD_3_OD): *δ* 2.88 (s, 3H, C*H*_3_N), 2.98–3.16 (m, 2H, C*H*_2_CH_2_N), 3.38–3.58 (m, 2H, NC*H*_2_CH_2_), 4.78 (d, *J* = 13.5 Hz, 1H, C*H*HN), 5.06 (d, *J* = 13.5 Hz, 1H, CH*H*N), 5.92 (s, 2H, OC*H*_2_O), 6.72 (dd, *J* = 8.1, 1.5 Hz, 1H, Ar *H*C-6), 6.75 (s, 1H, Ar *H*C-4), 6.77 (apparent bt, 1H, Ar *H*C-7), 7.56–7.65 (m, 2H, Ar *H*), 7.66–7.74 (m, 1H, Ar *H*), 7.78 (d, *J* = 6.5 Hz, 1H, Ar *H*), 8.01 (d, *J* = 9.0 Hz, 1H, Ar *H*), 8.06 (d, *J* = 9.6 Hz, 1H, Ar *H*), 8.21 (d, *J* = 8.8 Hz, 1H, Ar *H*); ^13^C NMR (125 MHz, CD_3_OD) *δ* 29.8 (1C), 39.4 (1C), 56.3 (1C), 57.4 (1C), 101.1 (1C), 108.1 (1C), 108.6 (1C), 121.6 (1C), 122.6 (1C), 125.1 (1C), 125.5 (1C), 126.3 (1C), 127.4 (1C), 128.9 (1C), 129.4 (1C), 130.9 (2C), 131.9 (1C), 134.1 (1C), 146.9 (1C), 148.1 (1C); Anal. Calcd for C_21_H_21_NO_2_.HCl: C, 70.88; H, 6.23; N, 3.94; Found: C, 70.48; H, 6.28; N, 3.92.

### 4.2. Molecular Docking Analysis

The putative binding sites of berberine analogues on FtsZ protein were determined through molecular docking analysis. The FtsZ protein was modelled using the I-TASSER Server [32], and the berberine structure was obtained using the LigParGen Server exploiting the Isomeric SMILES Code. The protein–ligand model was constructed using the PatchDock Server [33], and the structures were then refined with the FireDock Server [34]. All the interactions between the protein and ligand were determined using the Protein–Ligand Interaction Profiler (PLIP) Server [35]. Finally, the Gibbs free energy (ΔG) values were predicted using the PRODIGY webserver. All the figures were generated through UCSF CHIMERA software [36].

### 4.3. Experimental Assays

The pET-28a plasmid was used for overexpression of FtsZ in *E. coli*, and the protein was purified as previously described [21]. The purity of the protein was analysed by SDS-PAGE 12.5%, and its primary structure was validated by the MALDI mapping strategy on a 5800 MALDI-TOF/TOF instrument (ABI Sciex, Foster City, CA, USA).

The FtsZ GTPase assay was performed using 6 μM FtsZ in reaction buffer 25 mM PIPES/NaOH, pH 6.8. The mixture containing FtsZ was incubated for 30 min at 30 °C, and different concentrations of GTP (from 0 µM to 500 µM) were added for a reaction time of 10 min, either in the absence or in the presence of 40 µM **1c**, **1d**, **1f**, **1i**, **1g**, **1h**, **2g**, and **2h**. The reaction was performed for 10 min and stopped by the addition of 100 µL BIOMOL Green reagent. The amount of Pi released during the assembly of FtsZ was monitored using BIOMOL Green phosphate reagent (Biomol, Milan, Italy) after incubation at 25 °C for 25 min by measuring the absorbance at 620 nm. The negative control (without enzyme) was subtracted from all the readings. The experiment was performed in duplicate. The data were elaborated in order to report U/mg VS substrate concentration, and kinetic parameters were fitted by nonlinear regression with GraphPad Prism 8 (GraphPad Software, San Diego, CA, USA). The inhibition constant (*K*_i_) for all competitive inhibitors was calculated performing different kinetic analyses in the presence of 20, 40, and 60 µM of compounds and using the following equation:
$$K_{i}=\frac{\left[I\right]}{\alpha -1}\;\mathrm{where}\;\alpha =\frac{K_{M}^{app}}{K_{M}}$$

### 4.4. FtsZ Polymerization Assays

Assays for FtsZ polymerization by sedimentation were performed by incubating 6 μM FtsZ with 150 µM GTP in 25 mM PIPES/NaOH, pH 6.8 for 60 min at 25 °C. The assays were carried out in the presence and in the absence of 20 μM, 40 μM, and 60 μM of berberine analogues. The reaction mixtures were centrifuged for 60 min at 14,000 rpm, the pellets were run on 12.5% sodium dodecyl sulphate (SDS)-polyacrylamide gels, and stained by Coomassie blue. Subsequently, the images of the gels were acquired by ChemiDocTM, and the relative amount of FtsZ in the bands was determined densitometrically using the Quantity One software [10].

### 4.5. TEM Microscopy Analysis

FtsZ polymerization reaction solutions were diluted 1:10 in polymerization buffer in the presence and in the absence of 60 µM **1c** and **1i**. Then, 5 µL drops of the samples for TEM analysis were placed on a carbon-coated copper TEM grid for 20 min. Excess solution was wicked away with filter paper. Negative staining with 1% phosphotungstic acid (PTA) solution at pH 7.0 was carried out by depositing a drop of PTA solution on the grid for 1 min and then blotted dry with filter paper [37]. The grid was allowed to dry, and images were collected using a FEI TECNAI G2 S-twin apparatus operating at 120 kV (LaB_6_ source).

### 4.6. Fluorescence Spectroscopy

A 1.5 mL solution containing the **1c** compound was titrated by successive additions of FtsZ protein (from 0.010 to 0.120 µM). Titrations were done manually, and fluorescence spectra were measured in the range of 280–500 nm at the excitation wavelength of 295 nm. The fluorescence spectra were recorded using a Horiba Scientific Fluoromax-4 spectrofluorometer, using a quartz cell with an optical path of 1 cm, under controlled temperature conditions (Peltier control system at 20 °C). All experiments were carried out in duplicate. The change in the fluorescence intensity of the reaction set was fitted into the “one site-specific binding” equation of GraphPad Prism 8 (GraphPad Software, San Diego, CA, USA).

### 4.7. Determination of the Minimum Inhibitory Concentration Values of Berberine Analogues

The *E. coli* cells were grown in the presence of the serial dilution of berberine analogues from 500 μM to 0.5 μM, and the MIC values were determined as the lowest concentration showing no visible growth after 24 h of incubation at 37 °C. In brief, the bacterial strain was incubated overnight in LB at 37 °C. The cultures were diluted approximately to 5 × 10^5^ CFU/mL, and 50 μL of bacterial suspension was added to ten wells and incubated with serial dilutions of berberine analogues from an initial concentration of 500 µM. The sterility control well contained 100 μL of LB, while the growth control well contained 100 μL of microbial suspension. The plates were incubated at 37 °C for 16/20 h grown, and the MIC was determined by the lowest concentration showing no visible growth by measuring the absorbance at 600 nm. All determinations of all assays were performed in triplicate.

## Data Availability

Not applicable.

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
