# Peer review of "A Comparative Study of the Inhibitory Action of Berberine Derivatives on the Recombinant Protein FtsZ of E. coli"

_ijms, 2023, doi:10.3390/ijms24065674_

Round 1
Reviewer 1 Report
1. In Scheme 1 and 2, the position of R1, R2 and R3 should follow the same order.
2. In Scheme 1, a berberine’s structure should be added to compare with the analogues’ structure.
3. In line 120, 2h does not have a purine ring, it should be a naphthalene ring.
4. In Table 1, the author list the main interactions predicted for berberine analogues 1g, 1h, 1i, 2g, and 2h. For 1g, 1h and 2h, they all dock into the GTP binding site as the author claims. But there is a great difference in the interaction residues between each other, especially 2h. It has totally different hydrogen bonds interaction residues from that of 1g and 1h. for example, the hydrogen bonds of 1g and 1h come from (Gly19), Gly21 and Asn165, while 2h comes from Ala70, Ala72, Gly106, Gly107 and Thr108. The author should explain this difference exhaustively.
5. Figure 1 shows the docked binding site of different analogues on FtsZ. This is not sufficient for a docking result presentation. The detailed compounds’ conformation and interactions on the pockets should be illustrated, to verify the protein-ligand interaction analysis of Table 1.
6. Table 2 should be revised to make it easier to understand. KM should be indicated as KM for GTP; Vmax should be indicated as Vmax for FtsZ to hydrolyze GTP; Ki should be indicated as Ki for compound. More specific subjects in the table are shown below.
|
.. |
|||
|
With or without compound |
KM (μM) for GTP |
Vmax (U min-1) for GTPase |
Ki (μM) for compound |
|
/ |
50 ± 5 |
.. |
.. |
|
1c |
116 ± 5 |
.. |
.. |
|
1d |
88 ± 3 |
.. |
.. |
|
.. |
|
|
|
7. In section 2.3, the author writes in lines 156-158 “The GTPase activity of the FtsZ protein was assayed in the presence of the berberine related compounds at 40 μM concentration and was monitored in comparison with the untreated protein at different GTP concentrations. The kinetic parameters were calculated according to the Michaelis-Menten equation and the corresponding Lineweaver-Burk plots for all the berberine-related compounds. The results are reported in Table 2.” I’m wondering the Ki is calculated at a fixed compound concentration at 40 μM. Until to section 4.3, the author describes the Ki of Table 2 is calculated in a separate method, with 20, 40, and 60 μM of compounds. The description of the enzymatic activity, kinetic parameters, and the determination should be rearranged to avoid such misunderstanding.
8. In Table 2, the Ki of 1f, 1i, and 2g is missing. The author should explain this and add some signs in the box, like nd, in an annotation style.
9. Figure 2 does not show which is (a) and (b). And in the supposed (a), there are two 2g, one should be 2h.
10. In Section 2.4, the author writes “Among the berberine derivatives, 1c, f, i compounds seemed to exert the highest inhibitory effect as the amount of polymerized FtsZ was almost negligible when the inhibitor concentration was 60 µM.” That’s not fair, only compound 1i shows a negligible FtsZ polymerization in Figure 2(a).
11. In line 238, the description "in vivo" is not correct, it should be "in vitro".
12. In Section 2.6, the author tests the MIC of the compounds. For two newly synthesized compounds 2g and 2h, only 2g has a better MIC than berberine. And the author writes in line 246 “Data reported in Table 2 are in agreement with the hypotheses formulated in our previous study and are supported by kinetics, fluorescence spectroscopy, and docking experiments.” This description is too rough and indistinct. For example, 1g and 2h all dock into the GTP site, and also have similar enzyme kinetic parameters and FtsZ polymerization effect, but the MIC differs greatly (1g =128, 2h >512). And 1c, 1f, and 1i have the highest inhibitory effect on FtsZ polymerization (as the author writes in line 215), but their MICs are at distinct levels (1c = 128, 1f >512, 1i =256). These data is hard to be called "in agreement with" and "are supported by". The author should explain what he writes, by carefully and efficiently analyzing the MIC, kinetics, fluorescence and docking results in a head-to-head comparison manner between those compounds.
13. In line 289, the author describes compound 2g show "an acompetitive mechanism". What’s the difference of acompetitive from competitive and non-competitive?
14. In lines 309-311, the author writes “Among the analogues interacting with the active site, the presence of a small methyl group on the nitrogen atom leads to a remarkable increase in both inhibitory and antibacterial activities.” This description is not fair and overstate. In comparison between 1h and 2h, the methyl group on 2h does not increase antibacterial activity, conversely, 2h loses the antibacterial activity, (MIC of 1h is 128; 2h >512).
15. The author declares that “2g binds FtsZ in a totally different region compared to the other analogues” in line 314. This conclusion is too absolute, an in-silico docking result does not mean the real world.
16. In lines 318-321, “The data reported herein suggest that new berberine derivatives target the bacterial protein FtsZ demonstrating the potential of the berberine scaffold for chemical optimization in potent FtsZ inhibitors with broad-spectrum antibacterial activity and proposing a novel mechanism of action of new berberine analogues in inhibiting FtsZ.” The data presented in this manuscript does not support "a broad-spectrum antibacterial activity", the author just use one bacterial strain E. coli to test the MIC. And the claim "a novel mechanism of action" is not proved adequately and solidly.
17. The MIC unit (μg/mL) in Table 3 and the unit (μM) described in the determination Section 4.6 are different. It should be in the same unit.
18. FtsZ inhibitors will lead to bacteria enlargement or elongation on morphology. Such morphological change assay is suggested to be provided to further support the on-target FtsZ effect of these compounds.
Reviewer 2 Report
The manuscript “A comparative study of the inhibitory action of berberine derivatives on the recombinant protein FtsZ of E. coli” by Somma et al shows the effects of different berberine derivatives on FtsZ function and their antibacterial effects. The authors have synthesized different berberine derivatives and characterized their effects. Some of the derivatives shows better antibacterial effects compared to the parent compound. However, the experimental data do not supports the proposed mechanism or the claims. Authors need to explain several concerns in the manuscript.
The molecular docking predicts that berberine derivatives bind to different sites in FtsZ. However, this is just prediction. If the authors need to claim it, they need to perform mutations at respective sites to show a reduction in binding affinity for a particular derivative.
Did authors determined the number of binding sites of berberine in FtsZ? Does berberine binds to FtsZ non-specifically? How can authors explain it?
The GTPase activity of FtsZ is mostly represented as Pi release per FtsZ per min. The authors should provide this data, this will also help the readers to compare the effect of berberine derivatives with other anti-FtsZ compounds.
It is not clear how the competitive experiment was performed. Authors need to mention it clearly in the method section.
Table 2 title can be changed to more appropriate one, which should tell about the experimental findings.
Authors have concluded from the GTPase activity data that the mechanism are different for different analogues. Mere change in the binding or the GTPase activity will not tell so.
Sedimentation assay:
Authors have used only 150 uM of GTP and incubated the samples for 60 min. My concern is will FtsZ polymers stay that long at least for the control? If authors can determine the rate of ‘GTP hydrolysis per FtsZ per min’ that will show that longer incubation will lead to decrease in the polymer level.
Further, in the condition mentioned by the authors, FtsZ polymers require very high ×g value for sedimentation. 14000 rpm is quite low speed to sediment the polymers. Authors need to show both supernatant and the pellet samples side by side to show a decrease in polymer level. Further, authors need to show TEM images of FtsZ polymers, which will give higher confidence to the data.
The inhibition of FtsZ polymers is best in the case of ‘1i’. Is there is specific reason why it was not considered for further studies?
The binding of ‘1c’ with FtsZ was determined using fluorescence techniques with an excitation at 295 nm and emission at 327 nm. The berberine spectra shows that it has strong absorbance at both the wavelengths. Which means there will be a decrease in the intensity with increasing concentrations of berberine. Authors need to show the binding with at least one more technique.
Further, the Kd value is 26 nM. Why the authors are seeing the effects (both GTPase and polymerization) of berberine at 60 uM, which is almost 1000 times higher concentration?
Round 2
Reviewer 1 Report
none
Author Response
Thank you for the revision.
Reviewer 2 Report
Along with the Fig.1, the authors need to provide the docking scores (Del G values) for all the derivatives in a table.
Using in silico analysis, can the authors also show that ‘1i’ is not binding to GTP binding site and ‘1gh, 2h’ are not binding to C-terminal?
Did authors determined the number of binding sites of berberine in FtsZ? I asked this question to know if the compounds are binding non-specifically to FtsZ or not. To answer this query, the authors have referred to a previous manuscript doi: 10.3390/biomedicines9050452, however, it is also not mentioned there. Can the author’s perform Job’s plot to determine the number of binding sites of berberine in FtsZ?
In table 2, instead of ‘U/min’ please mention the data in ‘Pi released/min’
The quality of TEM images are poor and better TEM images should be included.
The absorbance spectra of 1c along with berberine needs to be included in the manuscript.
